# A Novel KRAS Antibody Highlights a Regulation Mechanism of Post-Translational Modifications of KRAS during Tumorigenesis

**DOI:** 10.3390/ijms21176361

**Published:** 2020-09-02

**Authors:** Mohamad Assi, Boris Pirlot, Vincent Stroobant, Jean-Paul Thissen, Patrick Jacquemin

**Affiliations:** 1de Duve Institute, Université Catholique de Louvain, 1200 Brussels, Belgium; 2Institut de Recherche Expérimentale et Clinique, Université catholique de Louvain, 1200 Brussels, Belgium; boris.pirlot@uclouvain.be (B.P.); jeanpaul.thissen@uclouvain.be (J.-P.T.); 3Ludwig Institute for Cancer Research, Université Catholique de Louvain, 1200 Brussels, Belgium; vincent.stroobant@bru.licr.org

**Keywords:** antibody, cancer, KRAS, prenylation

## Abstract

KRAS is a powerful oncogene responsible for the development of many cancers. Despite the great progress in understanding its function during the last decade, the study of KRAS expression, subcellular localization, and post-translational modifications remains technically challenging. Accordingly, many facets of KRAS biology are still unknown. Antibodies could be an effective and easy-to-use tool for in vitro and in vivo research on KRAS. Here, we generated a novel rabbit polyclonal antibody that allows immunolabeling of cells and tissues overexpressing KRAS. Cell transfection experiments with expression vectors for the members of the RAS family revealed a preferential specificity of this antibody for KRAS. In addition, KRAS was sensitively detected in a mouse tissue electroporated with an expression vector. Interestingly, our antibody was able to detect endogenous forms of unprenylated (immature) and prenylated (mature) KRAS in mouse organs. We found that KRAS prenylation was increased ex vivo and in vivo in a model of KRAS^G12D^-driven tumorigenesis, which was concomitant with an induction of expression of essential KRAS prenylation enzymes. Therefore, our tool helped us to put the light on new regulations of KRAS activation during cancer initiation. The use of this tool by the RAS community could contribute to discovering novel aspects of KRAS biology.

## 1. Introduction

The RAS family encompasses three genes that encode four related small GTPases, HRAS, NRAS, KRAS4A, and KRAS4B [1]. High sequence homology exists between the four proteins with significant divergences essentially present in the C-terminal tail. The *Kras* gene undergoes an alternative splicing giving rise to two isoforms, KRAS4A and KRAS4B [2,3]. The latter is the predominant variant and is extensively studied in cancer [4]. KRAS is mutated in 85% of all RAS-mutated cancers [5] and it plays a key role in the development of many aggressive malignancies, such as pancreatic cancer [6].

Cell membrane localization of KRAS is essential for its signaling activity [7]. Targeting KRAS to the cell membrane requires post-translational prenylation of its C-terminal tail. Prenylation consists in the addition of branched unsaturated lipid groups (palmitate and farnesyl groups for KRAS4A, and farnesyl group for KRAS4B) essential for membrane localization [8]. Three enzymes catalyze the C-terminal modifications of KRAS: farnesyltransferase (FTase), RAS-converting enzyme-1 (RCE1), and isoprenylcysteine carboxyl methyltransferase (ICMT). Pharmacological inhibition of FTase leads to the addition of a geranylgeranyl group by the geranylgeranyltransferase I (GGTase-I) [9].

Despite the remarkable progress that research has made on KRAS biology in the last thirty years, there is still a lack of tools to study KRAS expression, subcellular regulations, and post-translational modifications, especially in vivo. This directly limits our understanding of its biological regulations. In a general way, validated antibodies are efficient and easy-to-use tools that facilitate the characterization of protein functions and regulations. Unfortunately, many commercialized KRAS antibodies are of bad quality [10] because of inappropriate design and/or incomplete validation [11]. Recently, Water and coworkers found that among twenty-two commercialized RAS antibodies, only eight were able to detect RAS isoforms by Western blot but none of them were functional in immunolabeling [12]. In addition, these antibodies mainly recognize the endogenous prenylated form, and not the unprenylated form, of KRAS, which significantly limits the panel of research applications. In this work, we describe the first KRAS antibody that detects both unprenylated and prenylated forms of endogenous KRAS protein in mouse tissues. We discovered that the level of KRAS prenylation increases along with the protein expression of KRAS-prenylating enzymes in a model of pancreatic tumorigenesis. Finally, our antibody was validated for cell and tissue labeling.

## 2. Results

### 2.1. Design of Immunogenic KRAS Peptides and Production of KRAS Antibodies

To generate KRAS antibodies, a comparative analysis of sequence homologies between the human members of the RAS family was conducted to select peptide sequences specific to KRAS. Two regions were identified. In the first region, the sequence (peptide-1) was identical between KRAS4A and KRAS4B and differed by one and five amino acids from NRAS and HRAS sequences, respectively (Figure 1A). In the second region, the sequence (peptide-2) was specific to KRAS4B and differed by nine, nine, and ten amino acids from KRAS4A, HRAS, and NRAS, respectively (Figure 1B). The selected human KRAS peptides showed a perfect match with the corresponding mouse KRAS sequences (Figure 1A,B). In silico analysis showed that localization of the selected peptides in 3D conformation of KRAS should allow the antibody to access the corresponding regions (Appendix A). Each peptide was coupled to keyhole limpet hemocyanin (KLH) and injected into four rabbits following a classical immunization protocol. We tested the specificity of the different sera for KRAS by ELISA. Pre-immune sera showed no response (Figure 1C,D). For each serum, we observed a similar dose-dependent response after serial dilutions on wells coated with peptide-1 or peptide-2, indicating the presence of a specific anti-KRAS response (Figure 1C,D and Appendix A). Altogether, our results indicate that the produced polyclonal antibodies exhibit a specific activity for the immunized peptides.

### 2.2. Detection of KRAS in Transfected HEK-293 Cells and Electroporated Mouse Skeletal Muscles

To test the specificity of the produced antibodies, we then performed western blot analysis on protein lysates from HEK-293 cells overexpressing either the HRAS, KRAS, or NRAS protein fused to citrine (an improved version of GFP) [13]. The sera from eight rabbits were tested and their characteristics were listed in Table 1.

Antibody #5 and antibody #30 gave the best results and, therefore, will be further described in this paper. Pre-immune serum for antibody #5 failed to specifically recognize the different citrine-RAS proteins in HEK-293 lysates (Appendix A). For peptide-1, antibody #5 gave a strong and a weak signal for citrine-KRAS and citrine-NRAS, respectively, but did not cross-react with citrine-HRAS (Figure 2A). For peptide-2, antibody #31 recognized the three RAS proteins indifferently, and only antibody #30 specifically detected citrine-KRAS, albeit weakly (Figure 2A). A commercial pan-RAS antibody was used to confirm the presence of equivalent levels of citrine-HRAS, citrine-KRAS, and citrine-NRAS proteins in the HEK-293 lysates (Figure 2A).

Next, we assessed the functionality and specificity of antibodies #5 and #30 to recognize KRAS in immunolabeling experiments performed on HEK-293 cells overexpressing the citrine-RAS fusion proteins. Interestingly, for antibody #5, co-localization with direct citrine fluorescence produced by the citrine-KRAS fusion protein (green arrows) was seen; a weaker co-localization signal was also observed with citrine-NRAS (red arrows) (Figure 2B).

Antibody # 30 moderately co-localized with citrine-KRAS (orange arrows), but not with citrine-NRAS and citrine-HRAS (Figure 2C). Antibody #33, used here as a negative control, showed no signal with any of the citrine-RAS fusion proteins (Figure 2D). To test if the observations in HEK-293 cells could be confirmed on tissue sections, we electroporated mouse skeletal muscles with the different citrine-RAS expression plasmids. As expected, labeling with pre-immune serum on electroporated skeletal muscle sections did not give any specific signal for all of the citrine-RAS fused proteins (Figure 3A). In contrast, immunolabeling with antibody #5 showed a strong and weak co-localization with citrine-KRAS and citrine-NRAS, respectively (Figure 3A), which is similar to results obtained on HEK-293 cells. Quantification analysis of citrine-positive myotubes labeled with antibody #5 confirmed the greater sensitivity of this antibody for KRAS (89 ± 7.5%), compared to NRAS (23 ± 5.2%) (Figure 3B). Unfortunately, antibody #30 failed to give a clear signal for KRAS on skeletal muscle sections overexpressing citrine-KRAS. This is probably explained by the low titer of this serum and the unfavorable signal-to-background ratio resulting from its use at low dilution. Our results demonstrate that only antibody #5 can accurately detect KRAS on both cultured cells and tissue sections under overexpression conditions.

### 2.3. Detection of Unprenylated and Prenylated Forms of KRAS in Mouse Tissues

To broaden the experimental applications of antibody #5, we purified it by affinity chromatography. As depicted in Figure 4A, SDS-page micro-electrophoresis of the purified antibody #5 revealed the presence of the expected peaks corresponding to the light and heavy immunoglobulin chains. Next, we used purified antibody #5 to detect KRAS in adult mouse organs. Surprisingly, in addition to the expected band at 21 kDa, antibody #5 recognized a second band at a slightly higher molecular weight (Figure 4B). The upper band, of an unknown nature, was proportionally the most important in the pancreas while it was absent in the lung and spleen. As depicted in Figure 4C, a commercial KRAS antibody (WH0003845M1, Sigma Aldrich) only detected the lower band at 21 kDa; an identical result was obtained with two other commercialized KRAS antibodies (1415700 from Thermo Fisher and OP 24-100UG from Merck Chemicals). We hypothesized that these two bands represent the unprenylated (upper band) and prenylated (lower band) forms of KRAS, as such a lipid modification results in a migration shift on SDS-PAGE gels [14]. To test this hypothesis, we pharmacologically inhibited FTase and GGTase, two enzymes responsible for KRAS prenylation, in HEK-293 cells. In these cells, in the absence of treatment, only the lower band at 21 kDa was detected with antibody #5 (Figure 4D). Pharmacological treatment resulted in the appearance of an upper band, identical in size to that detected in pancreas lysates (Figure 4E), confirming our starting hypothesis. Altogether these results indicate that antibody #5 recognizes both unprenylated and prenylated KRAS forms in tissues.

### 2.4. Inflammation Promotes KRAS Prenylation

The above-mentioned finding suggests that KRAS is only partially active in the adult pancreas as it majorly expresses the unprenylated form; this form cannot reach the cell membrane and, thus, is not active [8]. As the pancreas is tolerant to the presence of an oncogenic *Kras* mutation and becomes sensitive when this mutation is associated with inflammation [15], we wondered if inflammation could increase the proportion of prenylated KRAS. To answer this question, we used *Ela-Kras^G12D/+^* mice in which we induced inflammation of the pancreas (pancreatitis) by treatment with cerulein. Interestingly, inflammation induced a migration shift reflecting an increase in KRAS prenylation (Figure 5A). Following cerulein treatment, 93% of KRAS proteins were prenylated compared to 54% in the control condition (Figure 5B).

To confirm this result, we used an ex vivo culture system that mimics the changes induced in the pancreas by inflammation [16,17]. In a similar way to a normal pancreas, on day 0 of culture, 60% of KRAS was unprenylated (Figure 5C,D). However, on day 3, a clear shift in KRAS migration was seen, the prenylated form became prominent to reach 75% of total KRAS (Figure 5C,D). The antibody that we generated revealed that inflammation of the pancreas leads to an increased KRAS processing, reflecting a higher proportion of the active KRAS form.

### 2.5. Pancreas Inflammation Promotes the Expression of KRAS-Prenylating Enzymes

The C-terminal processing of KRAS is controlled by three enzymes acting successively to increase its hydrophobicity and allow its targeting to the cell membrane. Our above result suggests that the expression of these enzymes is increased in the presence of pancreas inflammation. To test this hypothesis, we first measured the mRNA expression level of *FTase-α*, *FTase-β*, *Rce1*, and *Icmt* in normal and inflamed pancreata. We found a modest increase in the expression of these enzymes; only the expression of *FTase-β* showed a significant 2.5-fold increase (Figure 6A). In contrast, inflammation resulted in a sharp increase in the protein level of these enzymes, with respectively, a 6-, 3-, 7-, and 12-fold increase in the expression of FTase-α, FTase-β, RCE1, and ICMT (Figure 6B,C). These results suggest that inflammation increases KRAS prenylation by inducing the protein expression of the enzymes essential to this process.

### 2.6. KRAS Prenylation and Expression of KRAS-Prenylating Enzymes Correlate with the Susceptibility of the Pancreas to Initiate Carcinogenesis

Pancreata from newborns and young mice (1- to 15-postnatal days) are more prone to develop tumors than adults [15]. To test if the higher susceptibility of young pancreata to initiate tumorigenesis correlates with an increased prenylated form of KRAS, we measured the levels of KRAS prenylation, using our antibody. We found that pancreata from 10-day-old mice have 100% of KRAS in its prenylated form, compared to 60% in adult pancreata (60 days) (Figure 7A,B). This was associated with significantly higher expression levels of FTase-α and -β in young pancreata (Figure 7A–C); RCE1 and ICMT expression remained unchanged. Despite the very low expression of FTase-α/β, the prenylated KRAS form was still present in adult pancreas; this could be due to the presence of other enzymes important for KRAS prenylation, such as GGTase-I. Accordingly, we found that GGTase-I expression was similarly expressed in both young and adult pancreata, at easily detectable levels (Figure 7A–C). Consequently, a positive correlation between KRAS prenylation and the protein expression of FTase-α and -β was found (Figure 7D), suggesting that the latter positively influences the prenylation process of KRAS in the pancreas. Our data indicate that KRAS prenylation and the expression of related enzymes are associated with a higher susceptibility to initiate cancer.

## 3. Discussion

The lack of KRAS antibodies working in a large panel of applications is surprising because of its importance in the carcinogenesis process of many cancers; the reasons behind this lack are multiple. There is certainly an insufficient characterization of the commercialized RAS antibodies [12], which reflects a current global problem [18,19,20]. Another reason comes from the fact that KRAS shares high homology with HRAS and NRAS, which renders the generation of a specific KRAS antibody extremely challenging. In this work, we generated a sensitive KRAS antibody and demonstrated its utility by highlighting the presence of new regulations of KRAS during the initiation of pancreatic tumorigenesis.

RAS proteins exhibit 82 to 90% sequence similarity, except at the C-terminal tail that is the most divergent region [5,12]. In our study, peptide-2 from the C-terminal region of KRAS4B was weakly immunogenic in rabbits. Only 25% of rabbits engaged immune response that was lost over time. This explains why antibody #30 (peptide-2) gave a moderate signal for KRAS immunolabeling in cultured cells but failed on tissue sections. However, all rabbits injected with peptide-1 elucidated an adequate immune response. We found that antibody #5 (peptide-1) was able to immuno-detect KRAS with high sensitivity in cells and skeletal muscle tissues transfected with KRAS-expressing vectors. Although antibody #5 cross-reacted with NRAS, it showed greater sensitivity for KRAS; however, this cross-reactivity should be taken into account when interpreting experimental results. Another limitation of antibody #5 is the presence of non-specific background when immunolabeling is performed on mouse or human tissues expressing endogenous low-to-moderate levels of KRAS; thus, our antibody could be used for research but not diagnostic purposes.

Antibody #5 offers a much wider application range than that of the commercial antibodies currently available. Accordingly, our antibody could be used to easily detect the prenylation status of KRAS in healthy and diseased organs, by simply performing western blot experiments on protein lysates. Prenylation is an essential post-translational modification of KRAS that allows it to reach the cell membrane and activates signaling cascades [8]. For the moment, no commercialized antibody can sensitively detect endogenous levels of KRAS prenylation. The study of KRAS prenylation is usually limited to cell culture and is conditioned by the use of FTase and GGTase inhibitors. Therefore, studying the prenylation rates of KRAS in vivo is of particular interest in the cancer field. From this perspective, our antibody enabled us to emphasize an increase in KRAS prenylation rates in pancreas bearing *Kras^G12D^* mutation following the induction of pancreatitis. The high rates of KRAS prenylation were concomitant with an upregulation of three essential KRAS-prenylating enzymes, FTaseα/β, RCE1, and ICMT, whose expression seems to be controlled at the post-transcriptional level. These findings illustrate one of the mechanisms induced by pancreatitis to drive pancreatic tumorigenesis and provide an in vivo rationale to develop strategies targeting the process of KRAS prenylation. Therefore, efforts aiming to interfere with KRAS prenylation and target the cell membrane must be maintained and encouraged [21,22,23,24].

As shown in the present study, KRAS prenylation status in adult healthy organs is highly variable, suggesting that KRAS activity is differentially regulated between tissues. A comparative analysis of KRAS prenylation rates between healthy and diseased organs is helpful to better understand the dependency level of some organs on KRAS during the time course of disease. KRAS prenylation could be used as a biomarker to highlight the susceptibility of a given organ to initiate carcinogenesis in the presence of oncogenic mutations. In support of our interpretation, we reported here that young pancreata that are known to be more susceptible to cancer initiation than adult pancreata [15] express only the prenylated form of KRAS, contrary to adult pancreata, which also present unprenylated forms. From this perspective, the assessment of KRAS prenylation rates may provide important information on the progression of KRAS-driven cancers.

Some observations from our work could define the future strategy to improve the production of KRAS antibodies that optimally immuno-detect endogenous KRAS on tissue sections. The C-terminal region that is specific for each RAS isoform seems to be naturally less immunogenic compared with other RAS homologous regions, for unknown reasons. This means that injection of the full KRAS protein is expected to engage an immune response against the homologous regions between the different RAS proteins rather than the C-terminal tail. Accordingly, commercialized antibodies produced using full KRAS protein as immunogen are not effective and specific for KRAS immunolabeling in cultured cells [12]. Therefore, methods that differ from traditional immunization protocols must be privileged. In this context, the production of specific KRAS monoclonal antibodies by antibody phage display (APD) technique deserves to be tried. APD has been effective in producing antibodies recognizing specifically different isoforms of the Frizzled family sharing high homology, like the RAS family [25]. Our data could sensitize more biomedical scientists to work, validate, and produce efficient antibodies to allow novel reproducible discoveries.

## 4. Materials and Methods

### 4.1. Animals

Experiments on rabbits were performed in the animal facility of Eurogentec in accordance with the current ethical standards of the European Community (Directive 2010/63/EU). For transgenic mouse studies, we used *Elastase-Cre^ER^/LSLKras^G12D/+^* (*Ela-Kras^G12D/+^)* mice [6,26]. In these mice, *Kras^G12D^* mutation is induced in pancreatic acinar cells following gavage with tamoxifen (30 mg/mL in corn oil) and subcutaneous injection with 4-hydroxytamoxifen (0.3 mg/mL in corn oil). To induce acute pancreatitis, mice were treated with cerulein (125 µg/kg) for 5 days every other day (7 injections/day). Tissues were collected two days after the end of cerulein treatment. Experiments on mice were performed with the approval of the animal welfare committee of the Health Science Sector of UCLouvain (ethic approval number: 2017/UCL/MD/020, 17 August 2017).

### 4.2. Antibody Production

The selected peptide-1 and peptide-2 were synthetized on solid phase using Fmoc chemistry. HPLC analysis indicated that peptides have more than 90% of purity. Peptides were conjugated to KLH at a ratio 1:1 and then purified by desalting following manufacturer’s instructions (77606, ThermoFisher Scientific, Merelbeke, Belgium). For immunization, 200 µg of each peptide was subcutaneously injected with complete Freund’s adjuvant at a ratio of 1:1 into four rabbits. Immunization injections were performed at days 0, 14, 28, and 56. Rabbits were immunized following a classic program of 87 days. At day 0, pre-immune sera were collected before the first injection and was used as a negative control. Consecutive bleedings were performed at days 38, 66, and 87. The serum of the final bleeding of rabbit #5 was purified by affinity chromatography using the AF-Amino TOYOPEARL matrix conjugated to peptide-1 (40 mg of purified antibody from 11 mL of serum). After column washing with phosphate-buffered saline (PBS), antibodies bound to the matrix were eluted with glycine (100 mM, pH = 2.5). Antibody purity and activity were then evaluated.

### 4.3. Enzyme-Linked Immunosorbent Assay (ELISA)

Wells of a 96-well plate were coated with peptide-1 (15 µg/well) or peptide-2 (15 µg/well) and incubated overnight at 4 °C. Serially diluted sera were added to wells and incubated for 2 h at room temperature (RT). An anti-rabbit secondary antibody linked to horse peroxidase was added for 2 additional hours at RT. Finally, the o-phenylenediamine substrate was added for 20 min at RT, and color developed proportionally to the amount of KRAS antibodies. The reaction was stopped with H_2_SO_4_ (4 M) and optical density was measured at 492 nm using a microplate reader.

### 4.4. Cell Culture, Plasmid Transfection, and Pharmacological Treatments

HEK-293 cells were cultured in DMEM supplemented with 10% fetal bovine serum (FBS) and 1% penicillin-streptomycin at 37 °C and 5% CO_2_. For immunolabeling and Western blot, 15 × 10^4^ and 3 × 10^5^ cells were seeded in a 6-well plate, respectively. The next day, HEK-293 cells were transfected with a plasmid expressing a fusion protein for either citrine-KRAS, citrine-NRAS, or citrine-HRAS [13]. Transfection was performed using the CaCl_2_ method. A volume of 200 µL containing the plasmid (1 µg), CaCl_2_ (0.15 M), and HEPES-buffer solution (25 mM HEPES and 140 mM NaCl) were added dropwise on cells covered with 1.5 mL medium. After 24 h, the transfection medium was removed, and cells were used for Western blotting or immunolabeling experiments. For inhibition of FTase and GGTase, 3 × 10^5^ HEK-293 cells were seeded in a 6-well plate and treated the next day with FTI-277 (F9803-5MG, Sigma Aldrich, Overijse, Belgium) and GGTI-298 (G5169-5MG, Sigma Aldrich, Overijse, Belgium) at 50 µM and 15 µM, respectively, for 24 h. Controls were treated with 0.5% DMSO. The doses of FTI-277 and GGTI-298 inhibit around 50% of FTase and GGTase activity. The complete inhibition of FTase and GGTase leads to an extreme cellular mortality in HEK-293 cells, which limits the quantity and quality of protein lysates required for subsequent Western blot experiments.

### 4.5. Ex Vivo Culture of Dissociated Mouse Pancreas

Cultures of dissociated mouse pancreata that mimic inflammation-induced pancreatic acinar-to-ductal metaplasia were isolated as previously described [16]. Briefly, pancreata were cut into small pieces and digested with collagenase P (0.35 mg/mL in Hank’s balanced salt solution (HBSS) buffer). Collagenase incubation was performed for 15 min at 37 °C with 180 rotations per minute (rpm). The cell suspension was washed three times with HBSS/5% FBS and filtered through 500 µm and 100 µm strain filters. Finally, cells were dropped gently on a cushion of 30% FBS and centrifuged at 1000 rpm, for 2 min. The day of isolation (day 0) is equivalent to the situation in a normal pancreas. Metaplasia was spontaneously induced by maintaining pancreatic cells in 3D suspension culture, for 3 days, in Advanced RPMI medium supplemented with 5% FBS, 1% penicillin-streptomycin, and 0.1 mg/mL soybean trypsin inhibitor, at 37 °C and 5% CO_2_.

### 4.6. Skeletal Muscle Electroporation

Skeletal muscle electroporation was performed as described [27]. Mice with a CD1 genetic background were anesthetized using a ketamine (75 mg/kg) and xylazine (15 mg/kg) cocktail. Plasmids (1 µg/µL) expressing either citrine-KRAS, citrine-NRAS, or citrine-HRAS were injected transcutaneously in 10 different sites within the tibial anterior muscle (100 µL/muscle). Legs were shaved and a conductive gel was applied to ensure electrical contact. Transcutaneous pulses were applied by two stainless steel plate electrodes distanced by 1 cm. Ten pulses of 200 V/cm were administrated to each muscle with a delivery rate of 1 pulse/second. After 14 days, mice were sacrificed and muscles were harvested for immunolabeling.

### 4.7. Fluorescence-Activated Cell Sorting (FACS) and Real-Time Quantitative PCR (RTqPCR)

Acinar cells were isolated from pancreata of *Ela-Kras^G12D/+^* mice by FACS, as previously described [28]. RNA was extracted using a column-based approach (AM-1931, ThermoFisher Scientific, Merelbeke, Belgium). The RNA integrity number (RIN) was measured using Agilent 2100 Bioanalyzer and samples with RIN > 7 were selected for experiments. Reverse transcription was performed on 150 to 250 ng of total RNA using the M-MLV reverse transcriptase (M1705, Promega, Leiden, Netherlands). Quantitative PCR was carried out in a final volume of 10 µL (1 µL neosynthetized cDNA, 2 µL primers 10 µM, 5 µL Sybergreen KAPA mix 2× (KK4601, Sigma Aldrich, Overijse, Belgium), 2 µL nuclease-free water) using the CFX96 Real-Time System thermocycler (C1000, Biorad, Temse, Belgium). The expression of target genes was normalized to the *Rpl04* housekeeping gene. Relative expression was calculated using the ΔΔCt method. Primers used in the study are listed in Table 2.

### 4.8. Western Blotting

To obtain total protein extracts, tissues and cells were lysed in 50 mM Tris-HCl pH 7.4, 150 mM NaCl, 1% NP40 buffer. Protease (11836153001, Sigma Aldrich, Overijse, Belgium) and phosphatase (4906837001, Sigma Aldrich, Overijse, Belgium) inhibitors were freshly added just before lysis. Tissues were homogenized using Dounce homogenizer and cells were lysed by vortexing multiple times and gently pipetting up and down. Samples were maintained on ice during the procedure. Cell debris was pelleted after centrifugation (14,000× *g*, 10 min, 4 °C). Bradford protein assay (5000006, Biorad, Temse, Belgium) was used for protein quantification. Samples containing 20 to 50 µg of total proteins were electrophoresed on 7.5% to 12.5% SDS polyacrylamide gels. For the detection of KRAS prenylation, 12.5% polyacrylamide gels are needed. Protein migration and transfer were performed at constant voltage and amperage, respectively; for protein transfer, 120 mA/gel was applied for 90 min. PVDF membranes (ISEQ00010, Millipore, Brussels, Belgium) were blocked with a solution of 5% low-fat milk diluted in Tris-buffered saline (TBS)/0.1% Tween-20. Then, membranes were incubated overnight at 4 °C with the antibodies listed in Table 3. The next day, membranes were washed with TBS/0.1% Tween-20 and incubated with secondary antibodies for 1 h at RT. Images were taken using the Odyssey Imaging System (Li-COR, Lincolin, Homburg, Germany). HSC70 was used as a loading control. However, the expression level of HSC70 and other standard loading controls (i.e., β-Actin, GAPDH, α-Tubulin) was modified in the presence of inflammation. To solve this problem, when comparing normal and inflamed pancreata, we used Ponceau S staining to show that similar amounts of total protein were loaded. For the densitometry measurements of Ponceau S staining, the whole track for each sample was quantified using Image Studio Lite software, and then the ratio of target protein over Ponceau S staining was calculated.

### 4.9. Immunolabeling and Quantification

Dissected skeletal muscles were fixed in 4% paraformaldehyde (PFA) overnight at 4 °C, with gentle rotation, before paraffin embedding. Tissue sections of 6 µm were deparaffinized and antigen retrieval was performed in a microwave using citrate buffer (pH 6.0). Sections were washed with PBS once and then permeabilized with PBS/0.3% Triton-100X for 5 min at RT. Sections were blocked with solution 1 (3% low-fat milk, 5% bovine serum albumin, 0.03% Triton-100X in PBS) for 45 min at RT. GFP antibody (1/250; Ab6556, Abcam) or our homemade KRAS antibodies (1/200) were diluted in solution 1 and incubated overnight at 4 °C. The next day, slides were washed with PBS/0.1% Triton-100X and incubated with secondary antibodies diluted in solution 2 (10% bovine serum albumin, 0.3% Triton-100X) at 37 °C, for 1 h. Pictures were taken using Axiovert 200 microscope (Zeiss, Zaventem, Belgium). For cell labeling, transfected HEK-293 cells were washed once with PBS and then fixed with 4% PFA for 30 min at RT. Then cells were permeabilized, blocked with solution 1, and labeled with GFP (1/250) and homemade KRAS antibodies (1/200), as described above. Pictures were taken directly from the culture plate using the Axiovert 200 microscope (Zeiss). For myotubes quantifications, several pictures were taken for each muscle. First, the number of citrine-positive myotubes per picture was counted. Then, the number of myotubes labeled with antibody #5 was counted. Finally, the percentage of citrine-positive myotubes labeled with antibody #5 was calculated.

### 4.10. Statistical Analysis

Data were presented as means ± standard error of the mean (SEM). Normality and equal variance were checked before the statistical analysis was conducted. Comparisons between two groups were performed using an unpaired Student’s t-test. For all statistical analyses, the level of significance was set at *p* < 0.05. Analyses were performed using SigmaStat (version 3.1, Systat Software Inc., San Jose, CA, USA) and GraphPad Prism software (version 6, GraphPad Software, Inc., San Diego, CA, USA). * *p* < 0.05, ** *p* < 0.01, *** *p* < 0.001.

## 5. Conclusions

In this work, we generated and validated the first KRAS antibody labeling cells and tissues under overexpression conditions. This antibody was also able to sensitively detect unprenylated and prenylated forms of KRAS. Using our tool, we highlighted the presence of a new regulatory mechanism of KRAS activity during the initiation process of pancreatic tumorigenesis. This tool will offer new research applications to study KRAS biology and paves the way for novel discoveries in the cancer field.

## Figures and Tables

**Figure 1 ijms-21-06361-f001:**
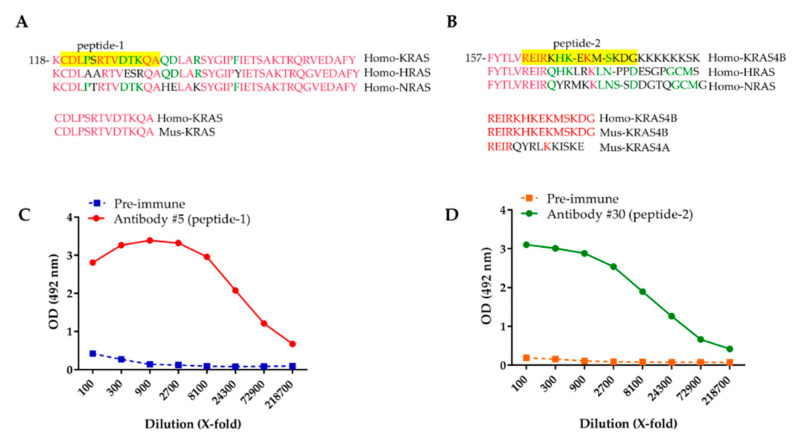
Generation of KRAS antibodies. (**A**,**B**) Partial sequence comparison between RAS family members. The peptide sequences (peptide-1 and peptide-2) used to immunize rabbits are highlighted in yellow. Peptide-1 and peptide-2 sequences are identical between human (Homo) and mouse (Mus) KRAS. Identical amino acids between all isoforms are in red. Identical amino acids between two isoforms are in green. Amino acids specific to one isoform are in black. The coordinate of the first amino acid of each sequence fragment is indicated. (**C**,**D**) ELISA with pre-immune sera, antibody #5, and antibody #30 on wells coated with peptide-1 and peptide-2, respectively.

**Figure 2 ijms-21-06361-f002:**
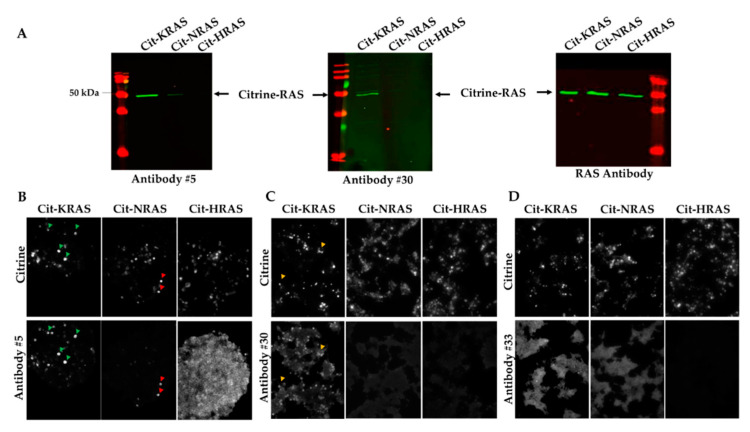
Validation of the specificity of KRAS antibodies by immunocytolabeling. (**A**) Western blots performed on HEK-293 overexpressing citrine-HRAS, citrine-KRAS, and citrine-NRAS using antibody #5, antibody #30, or a commercial RAS antibody. The citrine-RAS fusion proteins are detected at 48 kDa, as expected. (**B**–**D**) Immunolabeling performed on HEK-293 cells overexpressing citrine-HRAS, citrine-KRAS, and citrine-NRAS using antibody #5, antibody #30, and antibody #33. The latter is used as a negative control. The arrows show examples of cells where the direct citrine fluorescence co-localizes with the labeling of KRAS antibodies. Green arrows, co-localization of Cit-KRAS with antibody #5; red arrows, co-localization with Cit-NRAS with antibody #5; orange arrows, co-localization of Cit-KRAS with antibody #30. Cit: Citrine.

**Figure 3 ijms-21-06361-f003:**
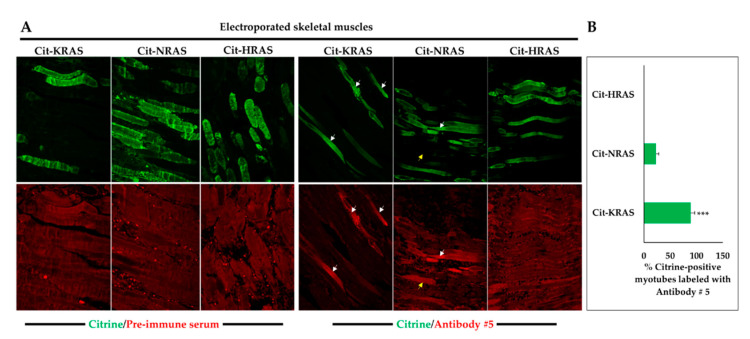
KRAS immunolabeling of electroporated mouse skeletal muscles. (**A**) Immunolabeling performed on electroporated mouse skeletal muscles with citrine-KRAS, citrine-NRAS, and citrine-HRAS (citrine labeling, green) using pre-immune sera (background, red) or antibody #5 (RAS labeling, red). White arrows show myotubes where citrine and KRAS labeling co-localizes. Antibody #5 also recognizes NRAS with lower affinity, as a higher background (yellow arrows) is observed to obtain a specific signal intensity equivalent to that obtained with KRAS. Cit: Citrine. (**B**) Percentage of myotubes co-labeled with citrine and KRAS or NRAS. Data are mean ± SEM. Statistical significance was tested using student’s *t*-test (***: *p* < 0.001).

**Figure 4 ijms-21-06361-f004:**
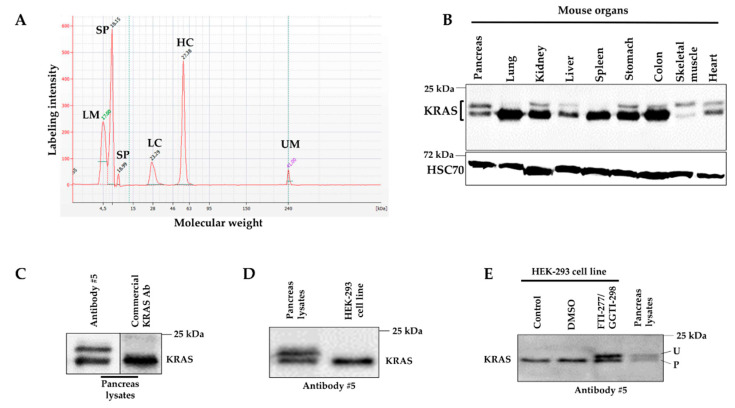
Detection of endogenous unprenylated and prenylated KRAS forms in mouse tissues. (**A**) SDS-page micro-electrophoresis showing the profile of purified antibody #5 after affinity chromatography purification. (**B**) Western blot performed on proteins extracted from mouse organs (*n* = 2). HSC70 is used as loading control. (**C**) Western blot performed on pancreas lysates using either antibody #5 or a commercial KRAS antibody (WH0003845M1, Sigma Aldrich) (*n* = 3). (**D**) Western blot performed on pancreas lysates and HEK-293 lysates using antibody #5 (*n* = 3). (**E**) Western blot performed using antibody #5 on lysates from the pancreas and HEK-293 cells which were not treated (control), DMSO-treated, and FTI-277/GGTI-298-treated (*n* = 3). U: unprenylated; P: prenylated. LM: lower marker; SP: system peak; LC: light chain; HC: heavy chain; UM: upper marker.

**Figure 5 ijms-21-06361-f005:**
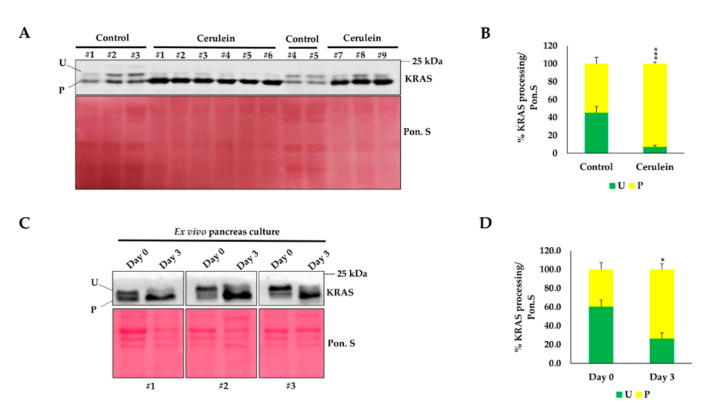
Impact of pancreas inflammation on KRAS prenylation. (**A**) Western blot on pancreas lysates from *Ela-Kras^G12D/+^* mice treated (*n* = 9) or not (*n* = 5) with cerulein for 1 week. This result shows the differential profiles of KRAS prenylation in the presence and absence of inflammation. (**B**) Densitometry quantification, from panel A, of unprenylated (green) and prenylated (yellow) KRAS forms normalized to Ponceau S (Pon.S), which is used as a loading control. (**C**) Western blot performed on protein lysates from ex vivo pancreas cultures at day 0 and day 3 using antibody #5 (*n* = 3). Day 0 is equivalent to a normal pancreas and day 3 is equivalent to a metaplastic pancreas, similar to that observed in the presence of inflammation. (**D**) Densitometry quantification from panel C showing the proportion of unprenylated (green) and prenylated (yellow) forms of KRAS. Normalization was performed with Pon.S. U: unprenylated; P: prenylated. Data are mean ± SEM. Statistical significance was tested using student’s *t*-test (*: *p* < 0.05; ***: *p* < 0.001)

**Figure 6 ijms-21-06361-f006:**
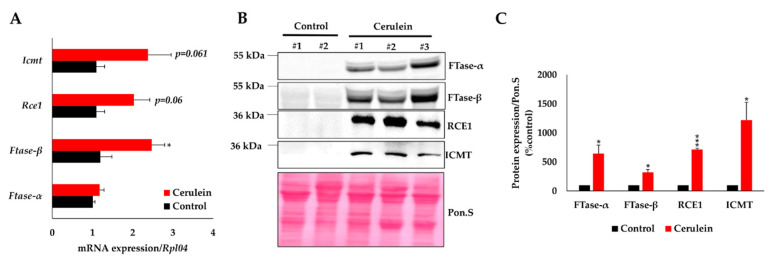
Pancreas inflammation induces protein expression of enzymes involved in the prenylation pathway. (**A**) RTqPCR analysis on FACS-sorted acinar cells from *Ela-Kras^G12D/+^* mice treated (*n* = 6) or not (*n* = 6) with cerulein for 3 days (*: *p* < 0.05). (**B**) Western blot analysis on pancreas lysates from *Ela-Kras^G12D/+^* mice treated (*n* = 3) or not (*n* = 3) with cerulein for 1 week. (**C**) Densitometry quantifications of blots shown in panel B. Data are mean ± SEM. Statistical significance was tested using student *t*-test (*: *p* < 0.05; ***: *p* < 0.001).

**Figure 7 ijms-21-06361-f007:**
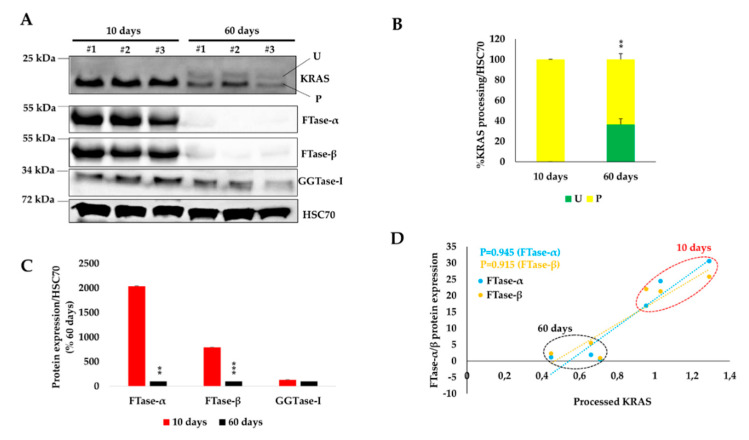
KRAS prenylation status in young and adult pancreata. (**A**) Western blot analysis on total pancreas lysates from young (10 days, *n* = 3) and adult mice (60 days, *n* = 3). (**B**) Densitometry quantification, from panel A, of unprenylated (green) and prenylated (yellow) KRAS forms normalized to HSC70, which is used as a loading control. (**C**) Densitometry quantification from panel A for FTase-α and –β, and GGTase-I. (**D**) Pearson’s correlation between prenylated KRAS form and the expression levels of FTase-α and -β in young and adult pancreata; densitometry values were used from blots shown in panel A. U: unprenylated; P: prenylated. Data are mean ± SEM. Statistical significance was tested using student’s *t*-test (**: *p* < 0.01; ***: *p* < 0.001).

**Table 1 ijms-21-06361-t001:** Characteristics of the different sera tested in this study. The level of target recognition is illustrated as follows: +++: high; ++: moderate; +: low. K: KRAS, N: NRAS, H: HRAS. #: hashtag precedes the specific number attributed for each rabbit.

	Applications	Rabbit #5	Rabbit #6	Rabbit #7	Rabbit #8
**Peptide-1**	Western blot	K (+++); N (+)	K (+++); N (+++); H (++)	K (+++); N (+)	K (+++); N (+++)
	Cell labeling	K (+++); N (+)	K (+++); N (++)	K (+); N (++)	K (+++); N (+++)
	Tissue labeling	K (+++); N (+)	K(+++); N (++)	None	K (+++); N (+++)
	**Applications**	**Rabbit #30**	**Rabbit #31**	**Rabbit #32**	**Rabbit #33**
**Peptide-2**	Western blot	K (+)	K (+++); N (+++); H (++)	None	None
	Cell labeling	K (+)	K (++)	None	None
	Tissue labeling	None	None	None	None

**Table 2 ijms-21-06361-t002:** List of primers used in the present study.

Gene	Forward (5′–3′)	Reverse (5′–3′)	Species
*FTase-α*	ATGGACGACGGGTTTCTGAG	TAAAGGCTCGTTCGCTCCTC	*Mus musculus*
*FTase-β*	TCCCCTGTTTGGTCAGAACC	GCATCCAGACACTCATAGGCA	*Mus musculus*
*Rce1*	GTGTCCTGGTAGTGTCCAGC	CAGGACAACCTTCAGCCCAT	*Mus musculus*
*Icmt*	CCGCCGGCTCTTCCG	AGCCAAGGAAACAAGCTCTGA	*Mus musculus*
*Rpl04*	CGCAACATCCCTGGTATTACT	TGTGCATGGGCAGGTTATAGT	*Mus musculus*

**Table 3 ijms-21-06361-t003:** List of antibodies used in Western blot experiments.

Antibody	Reference	Dilution	Antibody Incubation Condition
FTase-α	Sc-373749, SCBT	1/200	5% milk, overnight, 4 °C
FTase-β	Sc-46664, SCBT	1/200	5% milk, overnight, 4 °C
RCE1	Ab62531, Abcam	1/500	5% milk, overnight, 4 °C
ICMT	HPA032025-100UL, Sigma Aldrich	1/500	5% milk, overnight, 4 °C
RAS (EP1125Y)	Ab52939, Abcam	1/1000	5% milk, overnight, 4 °C
KRAS	WH0003845M1, Sigma Aldrich	1/500	5% milk, overnight, 4 °C
KRAS	OP24-100UG, Merck Chemicals	1/1000	5% milk, overnight, 4 °C
KRAS	415700, Thermo Fisher Scientific	1/1000	5% milk, overnight, 4 °C
Homemade KRAS antibodies		1/250	5% milk or BSA, overnight, 4 °C

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
