# Peer review of "A Novel KRAS Antibody Highlights a Regulation Mechanism of Post-Translational Modifications of KRAS during Tumorigenesis"

_ijms, 2020, doi:10.3390/ijms21176361_

Round 1

Reviewer 1 Report

 A novel KRAS antibody highlights a regulation  mechanism of post-translational modifications of  KRAS during tumorigenesis

In the manuscript the authors have developed a anti KRAS antibody with greater degree of specificity than the ones commercially available.

The manuscript requires more rigorous validation of their antibody.   Experiments with Isotype control is essential for all experiments. The following information are missing.

  1. It would be more conclusive to show In Silico analysis as a supplementary data.
  2. A detailed description of “Classical Immunization protocol is required.
  • How the peptides were prepared and purified.
  • What concentration was used?
  • Were any booster doses given? If yes at what concentration and interval.
  • How many times were the animals bled and sera collected?
  • What was the reason to choose rabbit over mouse when the immunogen is a short peptide?
  • A quantitative estimation of the volume of sera collected to the absolute quantity of affinity purified antibody is important for better understanding of the robustness of the immune response.
  • Four rabbits were used for each peptide (1 and 2) of which for peptide 1, two out of four rabbits (6 and 7) showed equal intensity for KRAS and NRAS while two had reduced affinity by western blot and cell labelling. This is a poor probability distribution and more animals should be included for statistical confidence regarding the specificity of the antibody.
  • Did all the four animals had same robust immune response and approximately same concentration of antibody was produced? It would be nice to show the immune response graphically.
  • Was the IgG subtype of the antibody determined?
  • An isotype control is essential in all the following experiments.

  1. In the mouse muscle electroporation experiment the sensitivity of the antibody# 5 is almost identical between KRAS and NRAS. How do the authors account for that? A graph with statistical significance after counting several slides from different animals will be more meaningful.
  2. Why was Ponceau S stained membrane used for normalization instead of regular housing keeping gene like HSC 70 or GAPDH or beta actin. If Ponceau S staining was used how exactly was the calculation done. Should be detailed.
  3. The source of the citrine-plasmids should be mentioned in the method with catalog number.
  4. It is relevant for the authors to discuss why they used affinity purified home made antibody at 1:250 dilution while the commercial was used at 1:500 and 1:1000 dilution.
  5. In figure 7a- authors show the lack of expression of FTase alpha and beta and but still the major proportion of the KRAS bands are prenylated in all three animals. How does the authors account for that?
  6. The discussion should emphasize more on the advantaged of determining the prenylated as well as the unprenylated form of KRAS, and why even after inhibiting the prenylation enzymes the prenylated bands are still observed.

Author Response

Dear Editor,
Please find the revised version of our ijms-904637 manuscript entitled “A novel KRAS antibody highlights a regulation mechanism of post-translational modifications of KRAS during tumorigenesis” by Assi et al.
We thank the reviewers for their constructive remarks that improved the quality of our manuscript. A point-by-point reply is provided on the submission platform of International Journal of Molecular Sciences. We have addressed all of their concerns, performed the required experiments and modified the manuscript accordingly. Modifications are highlighted in yellow throughout the text.
Looking forward to hearing from you soon.
Sincerely yours,
Mohamad Assi, PhD and Patrick Jacquemin, PhD

#Reviewer 1
In the manuscript the authors have developed a anti KRAS antibody with greater degree of specificity than the ones commercially available.
The manuscript requires more rigorous validation of their antibody. Experiments with Isotype control is essential for all experiments. The following information are missing.
1. It would be more conclusive to show In Silico analysis as a supplementary data.
As requested by the reviewer, In Silico analysis was added as Supplementary Figure S1.
This modification was cited in text: Results, paragraph 2.1., page 2, lines 74-75.
2. A detailed description of “Classical Immunization protocol is required.
 How the peptides were prepared and purified.
Peptides were synthetized on solid phase using Fmoc chemistry and characterized by mass spectrometry. HPLC analysis indicated purity of more than 90%. The peptide was coupled to KLH at 1:1 ratio and then purified by desalting. This was added in Materials and Methods, paragraph 4.2., page 11, lines 308-311.
 What concentration was used?
Rabbits received 200 μg of peptide per injection. This was added in Materials and Methods, paragraph 4.2., page 11, line 311.
 Were any booster doses given? If yes at what concentration and interval.
Yes, complete Freund’s adjuvant was used at a working concentration of 1:1 (adjuvant: peptide) and injected at days 0, 14, 28 and 56. This was also added in Materials and Methods, paragraph 4.2., page 11, lines 312-313.
 How many times were the animals bled and sera collected?
Bleeding was performed four times:
1-First bleeding before the first immunization (pre-immune sera were used as negative controls).
2-Second bleeding, 38 days after the first immunization.
3-Third bleeding, 66 days after the first immunization.
4-Final bleeding and sacrifice, 87 days after the first immunization.
Each rabbit was immunized four times at days 0, 14, 28 and 56.
These details were added to the text (Materials and Methods, paragraph 4.2., page 11, lines 313-316).
 What was the reason to choose rabbit over mouse when the immunogen is a short peptide?
The reason is that peptides elicit a weaker immunogenic response in mice than in rabbits (see for example (https://www.proteogenix.science/scientific-corner/antibody-production/antibody-production-rabbit-versus-mouse-which-host-best-fits-your-project/). This fits well with our experience as we tried to get mouse antibodies using the same peptides, but without success.
 A quantitative estimation of the volume of sera collected to the absolute quantity of affinity purified antibody is important for better understanding of the robustness of the immune response.
We obtained 40 mg of purified antibody from a serum volume of 11 ml. This was added in Materials and Methods, paragraph 4.2., page 11, line 317.
 Four rabbits were used for each peptide (1 and 2) of which for peptide 1, two out of four rabbits (6 and 7) showed equal intensity for KRAS and NRAS while two had reduced affinity by western blot and cell labelling. This is a poor probability distribution and more animals should be included for statistical confidence regarding the specificity of the antibody.
We did not attempt to give a statistical interpretation to the immunological responses elicited in the different rabbits used in our study; moreover, and in general, a statistical study is never coupled with the generation of a new antibody used for fundamental research purposes. Such a study, involving many other rabbits, would be very expensive and would take several months (which is incompatible with the revision process of this manuscript). We also respectfully believe that the scientific interest of the question is minor. For these various reasons, we did not respond positively to this specific point raised by the reviewer.
 Did all the four animals had same robust immune response and approximately same concentration of antibody was produced? It would be nice to show the immune response graphically.
The immune response of all the animals gave the same robustness following immunization with peptides. The data were added in Supplementary Figure S2.
This modification was cited in text: Results, paragraph 2.1., page 2, line 79.
 Was the IgG subtype of the antibody determined?
Rabbit has only one IgG subclass. For this reason, it was not necessary to determine the IgG subclass.
 An isotype control is essential in all the following experiments.
To our knowledge, isotype controls are generally used to validate monoclonal antibodies (see for examples: https://www.bio-rad-antibodies.com/ihc-controls-immunohistochemistry-tips.html and https://www.rndsystems.com/resources/protocols/importance-ihcicc-controls).
To best address the reviewer's concern about the specificity of the antibody, we included in our revised manuscript the use of pre-immune serum, which is frequently used as an experimental control (see for example "Controls for Immunohistochemistry", Hewitt et al, J Histochem Cytochem 2014, PMID 25023613). We added new western blots showing that pre-immune serum failed to detect specific bands and gave a high background in lysates from HEK-293 cells transfected with the different citrine-RAS plasmids (Supplementary Figure S3). Additionally, we added new pictures to Figure 3, showing the background signal of pre-immune serum on sections of electroporated skeletal muscles.
These modifications were also cited in the text: Results, paragraph 2.2., pages 3 and 5, lines 96-97 and 123-124.
3. In the mouse muscle electroporation experiment the sensitivity of the antibody# 5 is almost identical between KRAS and NRAS. How do the authors account for that? A graph with statistical significance after counting several slides from different animals will be more meaningful.
We believe that the sensitivity of antibody # 5 for KRAS is higher than for NRAS since the signal-to-background ratio is much better in favor of KRAS. This was explained in the legend of Figure 3, in the first version of our manuscript.
To confirm this, we performed quantification analysis, as requested by the reviewer. This convincingly shows that antibody #5 has a greater sensitivity to detect KRAS, compared to NRAS. These data were added to Figure 3 and to the text (Materials and Methods, paragraph 4.9, pages 13-14, lines 408-411 and Results, paragraph 2.2., page 5, lines 127-129).
4. Why was Ponceau S stained membrane used for normalization instead of regular housing keeping gene like HSC 70 or GAPDH or beta actin. If Ponceau S staining was used how exactly was the calculation done. Should be detailed.
The expression level of regular housekeeping genes like HSC70, GAPDH, β-Actin or α-Tubulin, varies markedly in the presence of inflammation. Therefore, the use of these standard loading controls is not optimal when comparing normal and inflamed pancreata. We used Ponceau S staining to show that similar amounts of proteins were electrophoresed for normal and inflamed pancreata. For the quantification of Ponceau S staining, the whole track of each sample was quantified using Image Studio Lite software, and then the ratio of target protein over Ponceau S staining was calculated.
As requested by the reviewer, these explanations were added to the text (Materials and Methods, paragraph 4.8., page 13, lines 388-393).
5. The source of the citrine-plasmids should be mentioned in the method with catalog number.
Citrine-RAS plasmids were a gift from Philippe Bastiaens, as mentioned in the Acknowledgments.
Reference 13, cited in the first version of our manuscript, corresponds to the study in which these plasmids were first developed and characterized.
6. It is relevant for the authors to discuss why they used affinity purified home made antibody at 1:250 dilution while the commercial was used at 1:500 and 1:1000 dilution.
The optimal dilution for each antibody was determined empirically. Importantly, the same result was obtained with commercial antibodies whether their dilution was 1:250, 1:500, or 1:1000. For reasons of purchase cost of commercial antibodies, we used the highest dilution giving an optimal result (1:500 or 1:1000).
7. In figure 7a- authors show the lack of expression of FTase alpha and beta and but still the major proportion of the KRAS bands are prenylated in all three animals. How does the authors account for that?
We thank the reviewer for this pertinent question. We think that despite the low expression of FTase-α/β in adult pancreas, other essential enzymes such as the geranylgeranyltransferase I (GGTase-I) could be expressed in normal adult pancreata and, thus, may explain the presence of the prenylated KRAS form. Accordingly, in the revised version, we tested the expression of GGTase-I in young and adult pancreata by western blot. We found that GGTase-I protein was clearly expressed at both stages, without any significant difference. This result suggests that the prenylated KRAS form could be due to the expression of GGTase-I. These new results were added to Figure 7A-B and were described in the text (Results, paragraph 2.6., pages 8, lines 220-224).
8. The discussion should emphasize more on the advantaged of determining the prenylated as well as the unprenylated form of KRAS, and why even after inhibiting the prenylation enzymes the prenylated bands are still observed.
As requested by the reviewer, we added a paragraph to discuss the usefulness of determining the rates of KRAS prenylation (Discussion, highlighted paragraph, page 10, lines 273-282).
The doses of inhibitors used in our study reduce the activity of prenylating enzymes by 50%. It is possible to achieve a 100% inhibition of prenylating enzymes by increasing the concentration of the used inhibitors, but this leads to extreme cellular toxicity and protein extracts are then of poor quality, incompatible with western blot experiments (Materials and Methods, paragraph 4.4, page 12, lines 338-341).

Reviewer 2 Report

The manuscript by Assi and colleagues refers to a new KRAS antibody and describe it as an improved tool to reach further comprehension on molecular mechanism attributed to these oncogene family. This manuscript sounds as relevant for scientific community.

I was wondering if the authors will commercialize these antibodies in the future.

The manuscript is clear and I don’t have any comments to do.

Author Response

Dear Editor,
Please find the revised version of our ijms-904637 manuscript entitled “A novel KRAS antibody highlights a regulation mechanism of post-translational modifications of KRAS during tumorigenesis” by Assi et al.
We thank the reviewers for their constructive remarks that improved the quality of our manuscript. A point-by-point reply is provided on the submission platform of International Journal of Molecular Sciences. We have addressed all of their concerns, performed the required experiments and modified the manuscript accordingly. Modifications are highlighted in yellow throughout the text.
Looking forward to hearing from you soon.
Sincerely yours,
Mohamad Assi, PhD and Patrick Jacquemin, PhD

#Reviewer 2
The manuscript by Assi and colleagues refers to a new KRAS antibody and describe it as an improved tool to reach further comprehension on molecular mechanism attributed to these oncogene family. This manuscript sounds as relevant for scientific community.
I was wondering if the authors will commercialize these antibodies in the future.
The manuscript is clear and I don’t have any comments to do.
We thank the reviewer for his/her nice comments.
Indeed, we have planned to commercialize the antibody and think to contact companies once the manuscript has been accepted.

Round 2

Reviewer 1 Report

The authors have tried their best to address all concerns. I data looks more conclusive now. I will recommend accept in the present form.

This manuscript is a resubmission of an earlier submission. The following is a list of the peer review reports and author responses from that submission.